# Towards Unification of General Relativity and Quantum Theory: Dendrogram Representation of the Event-Universe

**DOI:** 10.3390/e24020181

**Published:** 2022-01-25

**Authors:** Oded Shor, Felix Benninger, Andrei Khrennikov

**Affiliations:** 1Felsenstein Medical Research Center, Beilinson Hospital, Petach Tikva 4941492, Israel; shor.oded@gmail.com (O.S.); felixbenninger@gmail.com (F.B.); 2Sackler Faculty of Medicine, Tel Aviv University, Tel Aviv 6934206, Israel; 3Department of Neurology, Rabin Medical Center, Petach Tikva 4941492, Israel; 4Faculty of Technology, Department of Mathematics, Linnaeus University, 351 95 Växjö, Sweden

**Keywords:** general relativity, quantum theory, p-adic theoretical physics, dendrogram, event-universe, hierarchic structure, Mach’s principle

## Abstract

Following Smolin, we proceed to unification of general relativity and quantum theory by operating solely with events, i.e., without appealing to physical systems and space-time. The universe is modelled as a dendrogram (finite tree) expressing the hierarchic relations between events. This is the observational (epistemic) model; the ontic model is based on p-adic numbers (infinite trees). Hence, we use novel mathematics: not only space-time but even real numbers are not in use. Here, the p-adic space (which is zero-dimensional) serves as the base for the holographic image of the universe. In this way our theory is connected with p-adic physics; in particular, p-adic string theory and complex disordered systems (p-adic representation of the Parisi matrix for spin glasses). Our Dendrogramic-Holographic (DH) theory matches perfectly with the Mach’s principle and Brans–Dicke theory. We found a surprising informational interrelation between the fundamental constants, *h*, *c*, *G*, and their DH analogues, *h*(*D*), *c*(*D*), *G*(*D*). DH theory is part of Wheeler’s project on the information restructuring of physics. It is also a step towards the Unified Field theory. The universal potential *V* is nonlocal, but this is relational DH nonlocality. *V* can be coupled to the Bohm quantum potential by moving to the real representation. This coupling enhances the role of the Bohm potential.

## 1. Introduction

This paper is the first step towards the unification of *general relativity* (GR) and *quantum theory* (QT) on the basis of the Dendrogramic-Holographic (DH) theory [1,2]. The latter is based on a representation of the universe composed of events by dendrograms and at the ontic level by p-adic numbers [3]. Zero-dimensional p-adics serve as the basis of the holographic image. The event basis of physical theories was also used in the works of Wheeler [4], Smolin, and Barbour [5,6,7], and Rovelli [8]. (Systems are not present in DH theory).

DH theory can be considered as a part of the Wheeler’s “it from bit” project [4] on the information structuring of physics. However, our reconstruction is not as straightforward as Wheeler’s. The bit encoding is used for hierarchic relations between events in the universe described as dendrograms. Branches of dendrograms are strings of information.

The paper is not about “quantization of GR”. Following Smolin’s event-physics [5,6], we unify GR and QT on the basis of a new theory and mathematics, i.e., *the p-adic number system* [3]. Both classical and quantum random systems are represented by ensembles of dendrograms [1,2]. Quantumness is coupled to the simplicity of dendrograms. This simplicity characterization of quantumness is also the basis for Smolin’s theory. In contrast, we do not aim to reconstruct the traditional quantum formalism, but rather we use the test characterization of the quantum-like properties of dendrogram ensembles as via the CHSH test [2].

DH theory is also a step towards the creation of Unified Field theory. The events’ dynamics on dendrogram *D* are determined by the universal potential *V* representing hierarchical relations between events. By moving from the p-adic to real representation, *V* can be realized as Bohm’s quantum potential [1]. Thus, the latter can be considered as the universal (information) field. DH theory is nonlocal w.r.t. dendrogramic geometry, but this is informational nonlocality and not nonlocality in real space-time.

Now we will say a few words about p-adic theoretical physics. In 1987, a hypothesis of the possible p-adic structure of space-time was considered by Volovich [9]. This hypothesis was formulated in the framework of string theory, and it was supported by consideration of the p-adic analogue of the Veneziano amplitude and discussion on the main properties of the p-adic string theory. Since that paper, p-adic theoretical physics has been intensively developed [9,10,11,12,13,14,15] consistently with the development of the string theory ([9,16,17,18,19]) and complex disordered systems (The methodology of DH theory is like one used in ultrametric modeling of complex disordered systems: the aim is to find relational order which is not visible in the straightforward representation of data. Parisi and Sourlas [20] coupled this methodology to p-adic number theory and p-adic theoretical physics (see also Section 9)) [20] (p-adic representation of the Parisi matrix for spin glasses). One of the problems in this domain of science was the absence of coupling with real experimental data. In string studies, the p-adic structure of space-time was coupled to the Planck scale and the gap between this scale and the scale of the present physical experiment was too big; one could not expect direct experimental verification (the “usual string theory” over real has the same problem). We stress that one of the aims of p-adic theoretical physics was restructuring GR and cosmology [21]. This p-adic approach to GR was thus promising.

### 1.1. Dendrogram Coupling of P-Adics with Experimental Data

The natural question arises: *Can p-adic theoretical physics be somehow coupled to real experimental data?* (Additionally, without going to the deepest level of space-time, can it be coupled to the Planck scale?) Furthermore, can such coupling contribute to the development of gravitation theory? We answer these questions positively. The positive answer to the first question was given in papers [1,2] in the framework of *Dendrogramic-Holographic (DH) theory*. In this theory, experimental data, i.e., a time series, are represented (with some clustering algorithm, see Appendix A) by dendrograms, i.e., finite trees. Dendrograms represent hierarchically interrelated events: the event decomposition of the epistemic universe created by an observer *O*.

By increasing data collection, *O* can construct larger dendrograms. The limit of such dendrograms is the infinite tree. This is *the set of p-adic integers* Zp, and it represents *the ontic universe*. Hence, in DH theory, *p*-adic theoretical physics provides the ontic description of “reality as it is”. In principle, it can (but need not) be identified with the Planck scale physics. Thus, within DH theory p-adics are coupled with experimental data via dendrogram representation. We shall discuss DH theory in more detail in Section 2.

### 1.2. Dendrogram Representation of Geometry Corresponding to Metric Tensor

In GR, space-time geometry is determined by a metric tensor g = (g_{ij}). This geometry can be extracted from its corresponding geodesic equation and its solutions the geodesics. We can speak about the geodesic images of geometries: the *geodesic universe*. In this framework, the simplest “universe” is just a single geodesic. In DH theory, we start with such a geodesic universe. We then represent it by a dendrogram *D = D(g)* and get the *dendrogram universe*. Our basic tool is the extraction of data from the numerical simulation of photon trajectories on a space-time manifold. We consider DH images of batches of geodesics with discretized time and the corresponding *D* universes. As an illustrative model, we selected the Schwarzschild metric and considered batches of geodesics in a neighborhood of a black hole. In the *D* universe, we introduced analogues of the basic structures of “ordinary physics” (based on the real analysis) especially the analogues of the basic constants of nature.

The *D* universe is the hierarchic representation of the interrelation between events generated by (discrete) dynamics along geodesics. This is the *special information portrayal* of these events which is constructed by a clustering algorithm (generally, different algorithms create different *D* portrayals. However, as was shown in [2], DH theory is a stable w.r.t. selection of an algorithm).

For any DH universe, its treelike structure determines the universal potential *V_i*, depending on event *i* and branch *i* of *D*. This is the fundamental potential determining the hierarchic relations between events in *D*. (DH theory is about relations between events, not interactions between systems, cf. Wheeler [4]). In this paper, we consider the *D* universes constructed from batches of geodesics for the Schwarzschild metric (in a neighborhood of a black hole). Each *D* universe generates the universal potential $V_{i}=V_{i}\left(D\right)$.

### 1.3. Mach’s Postulate in the Dendrogram Representation

We now move to the DH theoretical approach to gravitation theory. We start with the basic coupling of DH theory with Mach’s principle [22]:

“[The] investigator must feel the need of… knowledge of the immediate connections, say, of the masses of the universe. There will hover before him as an ideal insight into the principles of the whole matter from which accelerated and inertial motions will result in the same way.”

In DH theory, this principle is not a postulate, but the fundamental property of the model. Representation of the event structure of the universe by dendrograms expressing the hierarchically ordered relations between events immediately leads to Mach’s principle. In a dendrogram, each point or branch of a tree is indivisibly coupled to all other points. This Machian constitution of DH theory is closely coupled to the non-mainstream pathway in the development of gravitation theory going back to Einstein’s paper (1911) [23], then Sciama (1953) [24], Dicke (1957) [25], and finalized in the Brans–Dicke theory of gravitation (1961) [26].

### 1.4. Dendrogram Counterparts of the Light Velocity and the Gravitational Constant

In the DH framework, *the light velocity c is not constant*. This is in the spirit of Einstein’s paper [23] in which *c* depends on the gravitational potential *Φ* (see also Sciama, Dicke, and Brans [24,25,26]). The gravitational constant *G* is not a constant either. In DH theory, each dendrogram-universe *D* is characterized by its own constants *c* and *G*, thus *c* = *c*(*D*) and *G* = *G*(*D*). They are coupled to the universal potential *V_i*.

By analyzing the dendrograms obtained from the batches of geodesics for the Schwarzschild metric (in a neighborhood of a black hole), we discovered the interrelation between *c*(*D*) and *G* = *G*(*D*): the fraction of their logarithms is approximately constant w.r.t. *D* corresponding to a variety of different batches of geodesics. The most striking is that this log-fraction approximately coincides with the corresponding log- fraction for physical constants *c* and *G*.

Some heuristics beyond this rate coincidence is that $log_{2}\;n$ gives the number of digits in the 2-adic expansion of a natural number *n*. This can be treated as an information measure. Thus, we found the interrelation between the information encoded in *c*(*D*) and *G*(*D*). To couple this interrelation to real physical constants, the latter should also be interpreted as information quantities as completed by Wheeler [24].

We will summarize the above discussion. We constructed dendrograms for batches of geodesics corresponding to the metric tensor. These geodesics and, hence, dendrograms, carry information about the basic constants of nature. We invented *D* analogues of these constants reflecting information encoded in real physical constants.

### 1.5. Dendrogram Approach to Quantumness

One of the aims of the DH theory project is the creation of a theory of quantum gravity through the unification of quantum and classical theories (So, we do not plan to quantize classical gravitation theory nor to create gravitation theory with the mathematical formalism of quantum field theory). Some steps towards this unification were taken in our previous papers [1,2]. In DH theory, quantumness and classicality are not sharply separated. Quantum events are represented by simpler dendrograms; complex dendrograms can be treated classically (cf. Smolin [6]).

In the present article, we combine the new mathematical formalism with a new scientific methodology. In contrast to the previous works, we do not try quantizing the classical theory or reducing quantum to classical. These are as two faces of Janus: the quantum face is visible in the rough representation of phenomena; collecting of more data makes visible the second face, the classical one. As was already mentioned, the general methodology, “simple systems are quantum and complex ones are classical” was presented by Smolin [6]. In contrast to us, he still (as is traditional in physics) used the language of systems, not events. In addition, our invention is the use of the new mathematical representation based on the treelike geometry corresponding to the hierarchic structure of relations between events composing phenomena. In the limit of infinite complexity this approach leads to the p-adic model of the universe.

The mathematical background for this model was developed in the works on p-adic mathematical physics [9,10,11,12,13,14,15,16,17,18,19,20]. However, in contrast to it, in DH theory p-adic points are not points of a kind of space-time (say space-time at the Planck scale) [9,10,11,12,13,14,15,16,17,18,19], but all possible events which would happen in the universe. This p-adic (ontic) universe is classical. The latter does not match the ideology of the previous p-adic physical modelling which considered “p-adic” as synonym to “quantum”.

In DH theory, classical and quantum physics are distinguished with the aid of experimental tests (Popper-like ideology) (A similar approach is basic for the theory of randomness (going back to the works of von Mises, Kolmogorov, Martin-Löf): randomness of concrete data series is checked with the aid of a batch of tests (say the NIST tests)). We completed the important step in this direction in our CHSH test paper [2], namely, it was shown that for relatively simple dendrograms (which are treated as representing quantum-like events) the Bell type inequality is violated. However, an increase in the dendrogram’s complexity implies a decrease in the degree of violation of the CHSH inequality. Of course, we demonstrated this transition from quantumness to classicality only for one test distinguishing statistics of quantum and classical events. This is the preliminary step towards justification of our methodology for quantum–classical unification.

The same DH methodology for quantum–classical unification can be applied to general relativity. We consider geodesics corresponding to some concrete metric as representing the relational structure of events. The representing of geodesics by simple dendrograms corresponds to the extraction of the quantum-like structure; representing them by complex dendrograms is classical(-like). For the moment, we have not yet elaborated a statistical test related to this situation. We can speculate about a kind of Bell test (or probability interference test) for dendrograms representing geodesics. However, the realization of the corresponding simulation is a complex task and we postpone it to a future paper.

In quantum mechanics, the Planck constant quantifies the irreducible uncertainty in the form of the Heisenberg relation. We also quantified the irreducible uncertainty of the dendrogram representation by introducing an analogue of the Planck constant for a dendrogram *D*, *h* = *h*(*D*). Surprisingly, we found the log-fraction interconnection between the dendrogramic light velocity *c* = *c*(*D*) and the Planck constant *h* = *h*(*D*). It is approximately constant for varying *D* (corresponding to batches of geodesics in the Schwarzschild metric). Moreover, as in the case of *c* and *G*, the dendrogramic log-fraction coincides with the log-fraction for the physical constants *c* and *h*. In such a consideration it is natural to use the information interpretation of the physical Planck constant *h* (see again Wheeler in [4]).

### 1.6. It from Bit: Dendrogram Realization of Wheeler’s Program

In his celebrated article [4], Wheeler presented the detailed program of the information reinterpretation of physics and in particular GR (In [4], Einstein’s geometrodynamics is mentioned a few times as the basic physical theory for the information reconstruction. In particular, Wheeler supported the “it from bit” idea by pointing out that “the surface area of the horizon of a black hole, rotating or not, measures the entropy of the blackhole”. In this discussion he appealed to the information interpretation of a quantum of action *h*): “**It from bit**. Otherwise put, every it—every particle, every field of force, even the spacetime continuum itself—derives its function, its meaning, its very existence entirely—even if in some contexts indirectly—from the apparatus-elicited answers to yes or no questions, binary choices, bits”.

We highlight the information viewpoint [4] on the Planck constant *h*: “The quantum, *h*, in whatever correct physics formula it appears, thus serves as a lamp. It lets us see horizon area as information lost, understand wave numbers of light as photon momentum, and think of field flux as bit-registered fringe shift. Giving us “its as bits”, the quantum presents us with physics as information”.

Wheeler’s critique of the mathematical models based on real numbers supports our move to dendrogamic and in the limit to p-adic physical models. See again [4]: “No continuum. No continuum in mathematics and therefore no continuum in physics”. The DH approach is based on a representation of physical reality (treated as reality of events in the universe) by dendrograms. These are treelike ordered sequences of bits, i.e., information strings.

## 2. DH Theory: Dendrogram Representation of Events and Zero-Dimensional P-Adic Holography

Here, we say more about DH theory (see [1] for the detailed presentation). Here, instead of physical systems, *an observer O forms events* by splitting experimental data into blocks and exposing these blocks to a hierarchic clustering algorithm (see Appendix A). For example, O divides the given time series of data into blocks of a fixed size *p*. In this way, *O* creates *the epistemic picture of the event-structure in the part of the universe, which is encoded in the data*, *O* universe (*Events and not physical systems, objects, are basic blocks of DH theory*. This viewpoint was strongly emphasized by Wheeler who cited the authors of [4]). If *p* > 1, then this dendrogram picture reflects the hierarchic relations between events. If *p* = 1, then this is the standard mapping of data on a real line. As was already mentioned, increasing the size of dendrograms (via collection of new data) pushes the limit where O approaches a p-adic model of ontic reality given by the infinite tree. This tree can be endowed with the algebraic structure, i.e., its (infinite) branches can be added, subtracted, and multiplied; this is the ring of p-adic integers denoted by the symbol ***Zp***. (If *p* > 1 is a prime number, then this ring can be extended to the field of the p-adic number ***Qp***). We remark that points of Zp (branches of the infinite tree) can be encoded by sequences of the form *x* = (*x*0, *x*1,…, *xn*,…), where *xj* = 0,1,…, *p* − 1, or series *x* = *Σ xj*
*p*^*j*.

The tree Zp is a metric space w.r.t. the metric *d*(*x*,*y*) = 1/*p*^*n*, where *xj* = *yj*, *j* = 0,…, *n* − 1, and *xn* ≠ *yn*. This metric can also be defined geometrically: take two branches *x* and *y* an find their common root, suppose it has the length *n*, then the distance between branches is given by the above formula. This is the so called *ultrametric*, i.e., the *strong triangle inequality* holds: in each triangle the third side is less than the maximum of the two other sides,
*d*(*x*,*y*) ≤ max [*d*(*x*,*y*), *d*(*y*,*z*)]

This inequality implies that in the ultrametric space *all triangles are isosceles.* Each ball in an ultrametric space is at the same time open and closed topogically (“*clopen*”), e.g., ball *B*(*a*,*r*) = {*x*: *d*(*x*,*a*) < *r*} is not only open (as one can expect), but also closed. Each point of a ball can be selected as its center. Two balls are either disjointed or one is contained in another. In particular, ***Zp*** can be represented as the disjointed union of *p* balls of the radius *r* = 1/*p*, each or the latter as the disjointed union of balls of the radius *r* = 1/*p*^2, and so on. This process generates disjointed partitions of ***Zp*** into *p*^*n* balls of decreasing radii, *r* = 1/*p*^*n*.

By starting from the ***Zp*** model of the universe (i.e., another way around), we create *the holographic representation of the epistemic universe*. The ***Zp*** is a zero-dimensional space, and it encodes the two-dimensional treelike geometry of dendrograms which in turn serve as codes for three-dimensional structures in Euclidean or Minkovsky geometries (Geometry of p-adic space exotic (comparing with Euclidean geometry). This is the *totally disordered and totally disconnected zero-dimensional topological space*. As was pointed out by Volovich [9], such geometry matches with heuristics on properties of geometry at the Planck scale. In [1,2], we also discussed matching with Bohm’s vision of implicate order [27]).

Now we discuss the process of creation of the dendrogram universe in more detail. An observer *O* “looks” at the universe (by using measurement devices of all kinds); *O* defines all unique events that he can discriminate. We say that the observer has for each event some epistemic level of discrimination. He constructs a finite dendrogram from the unique events at this epistemic level of discrimination. We call it the “universal dendrogram” of the observer. This is not the ontic dendrogram, which is infinite. Each event is represented on the epistemic dendrogram and encodes infinitely many ontic events that are indiscriminate in terms of the observer. Each branch of the dendrogram, a finite tree, encodes a ball in ***Zp*** containing infinitely many *p*-adic points, i.e., elementary ontic events. The *O* universe is described by relations between discernible events as the dendrogram shown in Figure 1A. Each event in the *O* universe is uniquely described by its p-adic expansion as in Figure 1B.

## 3. Universal Potential of Dendrogram

A dendrogram universe *D* is endowed with a potential and is denoted by *V_i*. In complete accordance with the Mach’s principle, this potential depends on the topology of whole *D*, i.e., it is a nonlocal function of *D*. It must be emphasized that in the implementations of DH theory to gravitation, we do not invent the metric tensor g = (g_{ij}); we operate solely with the *V_i*-potential. This is the *universal potential* determining all processes on *D* (which are in fact reduced to jumps between its branches or their endpoints on the bottom level of *D*). In this theory, all interactions are reduced to the universal nonlocal potential expressing topology of *D*.

For each edge (event) *i*, the p-adic expansion $V_{i}$ (where its computation is outlined in Appendix B step 5) represents the potential difference between the edge and the rest of the universe events. Thus, the sum of these $V_{i}$, these potential differences, is the non-referenced potential value V (This potential treated as a topological potential can be coupled to the quantum potential of Bohmian mechanics [1]. This coupling is not straightforward: one must move from the p-adic representation to the real one by using the so called Monna map based on p to 1/p transformation. Thus, in DH theory, the universal potential can be interpreted as a non-local Bohmian potential. However, this is merely heuristics and coupling between Bohmian mechanics, DH theory, and gravity is the complex problem. We shall work on it further). The potential difference between two edges (events) *i* and *j* is given by $V_{i}$ − $V_{j}$ = $q_{ij}$. The potential *V* represents the “difference between the observer and his universe”; symbolically, we can write O−universe=V. We can also symbolically write
$$\left(O-universe_{exept\;event\;i}\right)-\left(O-universe_{exept\;event\;j}\right)\;=V_{j}-V_{i}$$

We stress that the dendrogram picture is static, and dynamics can occur only upon jumping from edge to edge. Thus, the time-role or dynamical evolvement depends only on which event we jump to. In classical physics, based on the analysis on the real line, we have
$$\Delta p=F\Delta t\;\mathrm{and}\;W=\int_{x\left(t1\right)}^{x\left(t2\right)}Fdx=V\left(x\left(t2\right)\right)-V\left(x\left(t1\right)\right)$$

In the discrete case, the second Newton law (scaled to unit mass) has the form:$$\Delta p=F\Delta i\to F=\frac{\Delta p}{\Delta i}\;or\;(p_{j}-p_{i})/\left(j-i\right)=F$$
where $\int_{i}^{j}Fdi=V_{j}-V_{i}$ = p is the discrete differential.

$\left(V_{j}-V_{i}\right)/\left(j-i\right)=\Delta p$, where *j* − *i* = 1 is the minimal step of the discrete variable *i*.

We now turn to DH theory. In classical physics a trajectory *x(t)* is characterized by two variables, *x* for space and *t* for time. In our framework, these two variables are unified into one: the label i of the edge (or the end point of the dendrogram). This i can be represented either as a vector with 0/1 coordinates representing the path from tree’s root to the end-vertex lying at the bottom level of dendrogram *D* as a natural number. In our model, we define an analogue of the momentum only for a jump from event i to event j as the difference of potentials between these events, i.e., the quantity $V_{i}-V_{j}$ = $q_{ij}$ = $p_{i}-$
$p_{j}$.

We will summarize the above considerations. The topology of the dendrogram is described by the p-adic expansions of events encoded in the universal potential field given by the sequence of $V_{i}$. For each edge, its p-adic expansion $V_{i}$ represents the potential difference between the edge and the rest of the universe events. Thus, the sum of $V_{i}$, the potential differences, is the non-referenced potential value V. Importantly, *we do not have time and space coordinates* in DH theory. They are emergent quantities. Thus, we cannot clearly define the event’s momentum. Its role is played by the quantity $V_{i}-V_{j}$ = $q_{ij}$, i.e., the potential gap between two events represented by branches *i* and *j*. This potential gap, $q_{ij}$, can be considered as an analogue to the difference of kinetic energy from $edge_{i}$ to $edge_{j}$ and as outlined above as an analogue to delta momentum; $q_{ij}$ calculations are shown in Figure 1B.

## 4. From Mach’s Principle to Variability of the Basic “Constants” of Nature

In various studies of quantum gravity, the theory postulates Mach’s principle as first assumptions. In a very heuristic way, Mach’s principle states that the inertial forces acting on a body are a consequence of the quantity and distribution of matter in the universe.

On the other hand, DH theory does not need to postulate Mach’s principle. Mach’s principle is in fact a direct consequence of constructing a dendrogram. The dendrogram describes relations between matter objects in our observed universe or, in even more fundamentally relevant terminology, the dendrogram describes relations between our observational “events”. In DH theory, an event has no meaning without an observer and the rest of the observed universe (no dendrogram can be constructed in such a case, e.g., one observer and one event do not give rise to a dendrogram; we can consider the dendrogram only with an observer O and at least two events).

Attempts to follow the Machian perspective in constructing gravitational theory were made by Einstein already. Although the general theory of relativity has its Machian signatures, Einstein himself admitted he did not fully integrate Mach’s principle in the theory. An early study by Einstein [23] suggests that *the speed of light “in gravitational field is a function of place”* followed Mach’s principle very straightforwardly. Sciama [24], in 1953, developed a theory on the grounds of Mach’s principle that suggests *“inertia is not an intrinsic property of matter” but a consequence of matter relations*. Furthermore, his theory *“implies that the gravitational constant at any point is determined by the total gravitational potential at that point and so by the distribution of matter in the universe”* coupling local phenomena in the universe as a whole. We further note a study (based on Mach’s principle) by Dicke [25], where he formulated a gravitational theory with a changing speed of light as a function of relations to the whole universe matter distribution.

We start with the Einstein derivation [23] where he concluded that
(1)$$c=c_{0}\left(1+\frac{\Phi }{c^{2}}\right)$$
where $c_{0}$ is the speed of light at the coordinate origin c is the velocity of light at a given point with gravitational potential Φ.

However, Sciama’s derivation [24] suggested that
$$\frac{\Phi }{c^{2}}=-\frac{1}{G}\;where\;G=\frac{1}{\rho \tau^{2}}$$
where ρ is the density and τ is the Hubble onstant, i.e., the Hubble law has the form:
(2)v=τ R,
where v is the recessional velocity, typically expressed in km/s, and R is the proper from the galaxy to the observer *O* measured in mega parsecs (Mpc).

We also note that by Sciama
(3)$$\Phi =-\int \frac{\rho }{r}\;dV$$

Thus 1 and 2 give
(4)$$c=c_{0}\left(1+\rho \tau^{2}\right)$$
(5)$$\frac{c}{c_{0}}=\left(1+\rho \tau^{2}\right)\;\;\;and\;\;\;\frac{c}{c_{0}}=\left(1+\frac{\Phi }{c^{2}}\right)$$

Which results in
(6)$$\frac{\Phi }{c^{2}}=\rho \tau^{2}=\frac{c^{2}}{G}$$

We note that Sciama as well as Einstein derived these relations in a homogenous and isotropic distribution of matter of expending density ρ according to the Hubble law.

## 5. Constants of Nature as an Emergent Property of Dendrogram Topology

As was emphasized in the introduction, the dendrogram universe *D* is characterized by its own constants *c*, *G*, and *h*, so *c = c(D), G* = *G*(*D*), and *h* = *h*(*D*). These constants have the following surprising property: the log-fractions $\left|log_{2}c\;\left(D\right)\left|/\right|log_{2}h\;\left(D\right)\right|$ and $\left|log_{2}G\left(D\right)\right|/\left|log_{2}c\left(D\right)\right|$ are approximately constant w.r.t. *D* where dendrograms are generated by the clustering algorithm from batches of geodesics for the Schwarzschild metric in the neighborhood of a black hole (see Section 6 and Appendix B for the steps that produce dendrograms from geodesics).
(7)$$\left|log_{2}h\;\left(D\right)\right|/\left|log_{2}c\;\left(D\right)\right|\approx 3.91405517948,\;\mathrm{or}\;h\left(D\right)=c{\left(D\right)}^{\alpha },\alpha \approx 3.91405517948$$
(8)$$\left|log_{2}c^{2}\left(D\right)\right|/\left|log_{2}c\left(D\right)\right|\;\approx \;1.66610588966,\;\mathrm{or}\;c^{2}\left(D\right)=G{\left(D\right)}^{\beta },\;\;\;\;\beta \approx 1.66610588966$$

The log-quantities $log_{2}c\left(D\right)$, $log_{2}G\left(D\right)$, and $log_{2}h\left(D\right)$ give *the measure of information* contained in these numbers. Equations (7) and (8) express the stability of the fraction of the amount of information used for encoding these basic quantities. Our result is based solely on the numerical simulation. We hope that it will be supported by an analytic derivation in the future.

However, the most astonishing feature is the coupling of the dendrogram’s constants with the corresponding physical constants determined experimentally: *h*, *c*, and *G*. To formulate this coupling, we transfer these physical constants into the corresponding dimensionless quantities. Let us set a = 1 m^2^kg/s, u = 1 m/s, and g = 1 m^3^/kg s^2^. The quantities ***h*** = h/a, ***c*** = c/u, and ***G*** = G/g are dimensionless. We can now consider their logarithms $log_{2}h$, $log_{2}c$, $log_{2}G$, and find the fractions: (9)$$\left|log_{2}h\right|/\left|log_{2}c\right|=3.91405517948,$$
(10)$$\left|log_{2}c^{2}\right|/\left|log_{2}G\right|=1.66610588966$$

The coincidence of the LHSs of (7) and (9) as well as (8) and (10) is surprising. We cannot explain this coincidence theoretically; we interpret it as a sign that DH theory matches real physics.

We will define quantities *h*(*D*), *c*(*D*), and *G*(*D*) below.
$$\Phi \;will\;be\;attributed\;as\;the\;dendrogramic\;property\;V=\overset{n}{\underset{i=1}{\sum }}V_{i}\;as\;in\;\left(1\right)$$
$$We\;also\;introduce\;the\;dendrogramic\;quantity\;c\left(D\right)=\left(median\left(q_{ij}\right)\right)$$
i∈1,2…n−1 j∈i,i+1…n

$c\left(D\right)$ is calculated as follows: for each $V_{i}$ and $V_{j}$ such that i≠j we calculate the matrix
$$M_{i,j}=\;\left|V_{i}-V_{j}\right|\;=q_{ij}$$

Then the 50th percentile of the upper triangle values is $\left(median\left(q_{ij}\right)\right)$
$$G\left(D\right)\;is\;the\;dendrogramic\;property\;representing\;the\;mean\;function\;of\;density\;of\;events,\;\left(here\;we\;follow\;Sciama\right)\;2^{\alpha }*(\overset{n}{\underset{i=1}{\sum }}V_{i})/n\;where\;2^{\alpha }=1/\tau^{2}\;h\left(D\right)\;is\;the\;dendrogramic\;property\;representingthesquare\;of\;the\;information\;contained\;in\;the\;universal\;potential\;V\;h\left(D\right)=V^{2}$$
which gives the estimation of the indistinguishability present in Φ.

## 6. Dendrogram Representation of Geometry around Schwarzschild Black Hole

In order to construct a dendrogram representing some universe space-time, we decided to use a simulation of trajectories of photons, which represent geodesics on the space-time manifolds, emitted in the vicinity of a 2 + 1 Schwarzschild black hole. In order to simulate these trajectories, different local coordinates need to be connected by tensor networks.

We note that tensor networks in the proximity of a black hole have already been linked to p-adic numbers and p-adic trees with some analytical solutions that indicate the emergence of gravity, quantum field theory, and holography [27,28,29,30,31]. The Schwarzschild metric describes the gravitational field outside a spherical mass where the electric charge of the mass, angular momentum of the mass, and universal cosmological constant are all zero. The geodesics which were simulated by the Schwarzschild metric already include information on the connection of different local coordinates through tensor networks as well as information on the Schwarzschild black hole mass.

We emphasize that in DH theory all events of the universe are statically present with no dynamics. An apparent casual structure emerges upon maximizing a certain action principle (see Section 8). Thus, the black hole mass, which defines the space-time manifold geometry, defines also all possible events and thus all possible geodesics and casual structures.

Alternatively, we have to point out that the space-time geometry and the events it encompasses define the black hole mass.

The Schwarzschild metric is given by
$$ds^{2}=-\left(1-\frac{r_{s}}{r}\right)dt^{2}+{\left(1-\frac{r_{s}}{r}\right)}^{-1}dr^{2}+r^{2}d\Phi^{2}{{}^{}}^{}$$
where

c is the speed of light, r is the radial coordinate, $r_{s}$ is the Schwarzscild radius, and Φ is the longitude.

Thus, we produced three sets of geodesics’ events around a Schwarzschild black hole. These geodesics are formed by pulses of light at the Schwarzschild radius

*r* = [1.5 1.7 1.9 2.1 2.3 2.5 2.7 2.9] and

Φ = [0 π/2 π π × 3/2] in our “universe”

The output of the simulation is generated data sequences of [t x y] coordinates for each emitted photon geodesic.

*t* = [0,5] where each pulse is 10 photons: 761,600 total events from 320 geodesics.*t* = [0,10] where each pulse is 10 photons: 1,523,080 total events from 320 geodesics.*t* = [0,10] where each pulse is 20 photons: 3,046,160 total events from 640 geodesics.

We first analyzed each of the universal dendrograms constructed out of events in universes 1–3 (Figure 1C). Each such universal dendrogram was constructed in the following way:

Each geodesic in the universe was coarse-grained by a factor k (jumping from one event to the next kth event). Then, we constructed a universal dendrogram from all coarse-grained geodesic events. Figure 2A shows values for each of the universes (1–3), the $\left|log_{2}h\;\left(D\right)\right|/\left|log_{2}c\;\left(D\right)\right|$, and $\left|log_{2}c^{2}\left(D\right)\right|/\left|log_{2}G\left(D\right)\right|$ ratio compared to the same log-ratios of the physical constants determined experimentally upon increasing the size of the universal dendrogram. We note that in different units, the selection of the log-ratios of the physical constants determined experimentally had different values. We show that the scaling of regular units corresponded to power scaling of the full dendrogram. Thus, when we use the Kg-m-s units, we multiply each $V_{i}$ by 2^1^ while in the kg-cm-s ($\left|log_{2}h\right|/\left|log_{2}c\right|=2.7850761987,\;\left|log_{2}c^{2}\right|/\left|log_{2}G\right|=5.01810477316$) we multiply each $V_{i}$ by 2^20^.

We note that $G\left(D\right)$ is also coupled to the dendrogramic Hubble constant. This dendrogramic Hubble constant in our model is a free parameter that is adjusted according to size and topology of the dendrogram in order for the $\left|log_{2}c^{2}\left(D\right)\right|/\left|log_{2}G\left(D\right)\right|$ ratio to be in accordance with the experimental $\left|log_{2}c^{2}\right|/\left|log_{2}G\right|$ ratio. We show in Figure 2B the change of the factor $1/\tau^{2}$ (see Equation (2)) with the universal dendrogram size. This analysis suggests that the experimental h, c, and G are a consequence of the relational properties of our real universe. Thus, the Hubble constant is a relational property that is linked to the size and relational topology of our real universe. Other smaller but similar relational topological structural universes must scale the Hubble constant. Thus, it remains a free parameter in our formulation. Figure 2B shows how we scale the Hubble constant in order to match the experimental ratio $\left|log_{2}c^{2}\right|/\left|log_{2}G\right|$. Currently—until we find what property of the dendrogram this parameter represents—we are left with the one fundamental ratio |($\left|log_{2}h\left(D\right)\right|/\left|log_{2}c\left(D\right)\right|)$ that depends solely on the dendrogram topology.

Moreover, for each geodesic, we constructed its own geodesic universal (Figure 1C) dendrogram upon increasing k factor of coarse-graining. Figure 2C,D show the mean values of the log-ratios $\left|log_{2}h\left(D\right)\right|/\left|log_{2}c\left(D\right)\right|$ (Figure 2C) and $\left|log_{2}c^{2}\left(D\right)\right|/\left|log_{2}G\left(D\right)\right|$ (Figure 2D). These are less in agreement with the log-ratio of experimentally determined constants. For each k (k = 1,2…20), the log-ratios approach the real values of $\left|log_{2}h\right|/\left|log_{2}c\right|=3.91405517948,\;\left|log_{2}c^{2}\right|/\left|log_{2}G\right|=1.66610588966$ but with significantly less precision than obtained from the dendrograms constructed from all 320/640 geodesics (Figure 2C,D). Our coarse-grained universe is constructed from one geodesic, with a clearly different topology of the dendrogram (compared to a dendrogram constructed from several geodesics); we cannot agree with the experimentally tested constants h and c. It seems that we do need a universe with more then one geodesic (and probably *homogenous and isotropic distribution of events*) to agree with the experimentally and physically determined constants.

## 7. Geometrical Meaning of Constants of Dendrograms as Similarity Measures

We claim that these dendrogramic constants $h\left(D\right),\;c\left(D\right)$, and $G\left(D\right)$ are properties that measure how much one dendrogram is scale-free similar to another dendrogram. Hence these are measures of similarity between any two systems. In our model, one of the systems is the entire observable universe with informational properties manifested by the physical constants determined experimentally. The second system uses the little universes we created. Heuristically, we can envision a dendrogram as a triangle with a discrete base where $h\left(D\right),\;c\left(D\right)$, and $G\left(D\right)$ are scale-free properties of this triangle. Let us define these properties in a more geometrical way. See Figure 3.

Thus, when the ratio between two properties of a triangle is close to the same properties’ ratio in a different system, these triangles/universal dendrograms are proportional/similar. Our results suggest that if we could make a dendrogram from all the events of our universe, then this dendrogram would be proportional to the little universes that we created.

## 8. The Emergent P-Adic Path

Our aim in this section is to describe a dynamical process on the static universal dendrogram. (We remind the reader that the epistemic universal dendrogram, with no apparent dynamical process, is composed of all events in the real dynamical world of [t x y] coordinates). For that purpose, we again produced three coarse-grained universes: the first with 25,560 events composed out of 320 geodesics, the second with 50,920 events composed out of 320 geodesics, and the third with 67,920 events composed out of 640 geodesics. We also acquired much more detailed (30-fold more) data for each of the geodesics.

We follow the reasoning outlined in our previous study [1] and the D analogue of the action principle suggested by Smolin for the casual set theory; we consider it as phenomenological action, and thus:
$$V_{i}=\overset{n}{\underset{j=1}{\sum }}(a_{j}2^{j})$$
is the measure of distinguishability of edge *i* from all edges j≠i; we mention again that $V_{i}$ represents the potential difference between the edge event and the rest of the universe events. The sum of these $V_{i}$, these potential differences to the rest of the universe, is the non-referenced potential value V. The action of this potential field, SRE, is taken to be proportional to the potential value V:

$SRE=gV\;where\;g\;is\;a\;proportion\;constant,\;where\;V=\overset{n}{\underset{i=1}{\sum }}V_{i}$.
$$S^{ECS}+S^{RE}=\overset{n}{\underset{i=1\;j=i+1}{\sum }}\tilde{N}\frac{q_{ij}^{2}}{2}+\sum_{i=1}^{n}{\tilde{z}}_{i}P_{i}+gV$$

As shown in the results, $q_{ij}$ should be maximal through the chronological measurement/dynamic process. Thus:
$$P_{i}=q_{i,i+1}-q_{i+1,i+2}$$

The variation by $q_{ij}$ yield:
$$0=\tilde{N}q_{ij}+{\tilde{z}}_{i}+gV_{i}$$

As shown previously [6] in the casual set theory suggested by Smolin, the space-time coordinates are represented by the Lagrange multiplier ${\tilde{z}}_{i}$ after substituting $q_{ij}$ and $V_{i}$ into the equation above. Thus, the space-time intervals are the differences between ${\tilde{z}}_{i}$ and ${\tilde{z}}_{j}$. As $V_{i}$ and $q_{ij}$ are zero-dimensional p-adic numbers, ${\tilde{z}}_{i}$ are also zero-dimensional p-adic coordinate numbers that are an emergent consequence of the dendrogramic structure. We emphasize that the casual structure on the static dendrogram is an emergent property. This casual structure emerges upon maximizing the above action principle according to the dendrogramic structure (as will be demonstrated below in Section 8.1, Section 8.2, Section 8.3 and Section 8.4). this dendrogramic casual structure is in agreement with the “real world” casual structure. Moreover, although different coordinates’ systems, metrics, and linkage algorithms will produce different dendrogramic casual structures, the agreement with the “real world” casual structure will be preserved.

### 8.1. P-Adic Coordinates of Single Geodesic Dendrogram follow the Maximal Path

In accordance with the above action-variation, we further describe each geodesic in terms of the p-adic ${\tilde{z}}_{i}$ coordinates (or rather points) resulting in a p-adic path. GR indicates that the chronological path taken by a photon from its emitting point down to the last point in space-time is minimal in the Schwarzschild coordinates. For that purpose, we constructed a dendrogram for each geodesic. We showed that the path on such a dendrogram from edge_1_ to edge_last_ results in an emergent p-adic coordinate sequence ${\tilde{z}}_{1},{\tilde{z}}_{2},{\tilde{z}}_{3}\ldots {\tilde{z}}_{last}$ where the sum of $log_{2}\left|{\tilde{z}}_{i}–{\tilde{z}}_{i+1}\right|p$ will be maximal suggesting that ${\tilde{z}}_{i}$ and ${\tilde{z}}_{i+1}$ are p-adically closer and more similar.

This p-adic chronological path taken is in fact maximal in comparison to any other randomly chosen p-adic path from edge_1_ to edge_last_. For each geodesic, we chose 10,000 random edge-to-edge paths starting at the chronological edge_1_ and ending at the chronological edge_last_. We then calculated for the geodesic and the randomly selected 10,000 alternative paths whose ${\tilde{z}}_{i}$ coordinates by ${\tilde{z}}_{i}=-\tilde{N}q_{ij}-gV_{i}$ were g and $\tilde{N}=1\;\mathrm{and}\;i\in 1,2..n-1$. We calculated the intervals between consecutive ${\tilde{z}}_{i}$ as $ds=log_{2}\left|{\tilde{z}}_{i}–{\tilde{z}}_{i+1}\right|$ This is in fact a degree of similarity, ds between ${\tilde{z}}_{i}$ and ${\tilde{z}}_{i+1}$.

For the chronological sequence and the randomized scrambled geodesic

$path=\overset{n-1}{\underset{i=1}{\sum }}ds$. For each scrambled geodesic, we calculated the ratio

$path_{geodesic}/path_{random}$ where $path_{geodesic}$ was the geodesic from which the $path_{random}$ was scrambled. We plotted the CDFs of all such ratios in all three universes (Figure 4). As can be seen in Figure 4, all $path_{geodesic}/path_{random}$ values are above the value of one which means p-adically that the geodesic is the shortest possible path. For universes 1–3 the CDFs have the corresponding mean ± std values, 2.9479 ± 0.3653, 3.3113 ± 0.5305, and 3.296 ± 0.5084.

The CDFs showing all ratio $path_{geodesic}/path_{random}$ values are above 1 for every geodesic in all universes 1–3.

### 8.2. P-Adic Coordinates of Single Geodesic Edge as Part of the Universal Dendrogram follow a Maximal Path

We next verified that we see this effect also in a universe with 320/640 geodesics when all events from all the geodesics are clustered together into a dendrogram. The only difference is that now each geodesic’s edges are mixed with the other geodesics’ corresponding edges. We identified each geodesic edge in the big universal dendrogram and carried the same analysis as in Section 8.1 for the calculation of the ratio $path_{geodesic}/path_{random}$ (Figure 5). As can be seen in Figure 5, all $path_{geodesic}/path_{random}$ values are above the value of one which means p-adically that the geodesic is the shortest possible path. For universes 1–3 the CDFs have the corresponding mean ± std values, 2.3017 ± 0.5646, 2.3471 ± 0.3498, and 2.4355 ± 0.4192.

### 8.3. Geodesics as Sub-Universes

We tested whether an event from one geodesic will dynamically transfer to a different event in a different geodesic. Thus, we calculated the potential gap $q_{ij},\;where\;i\;j\;are\;edges\;that\;belongs\;to\;the\;same\;geodesic$, and the potential gap $q_{ir}\;where\;i\;are\;edges\;that\;belong\;to\;one\;geodesic\;and\;r\;belongs\;to\;another\;geodesic$.

We noticed that in the mean $log_{2}\left|q_{ir}\right|$, where
i are all edges of one geodesic and all r belong to another geodesic, these values are very distant from the mean of $log_{2}|q_{ij}{|}_{p}\;where$
i and j are all edges of one geodesic. Figure 6 indicates that on average two edges from two different geodesics cannot communicate because their p-adic potential gap is larger than the difference allowed by each of the geodesic’s mean potential gap. This can be seen by dividing the geodesic’s mean $log_{2}|q_{ij}{|}_{p}$ by each of the two geodesics’ mean $log_{2}\left|q_{ir}\right|$. When the resulting number is greater than one, this means p-adically that $q_{ir}$ is bigger than the average potential gap inside a geodesic (Figure 6). For universes 1–3 the CDFs of the ratio $mean\left(log_{2}|q_{ij}{|}_{p}\right)/mean\left(log_{2}|q_{ir}{|}_{p}\right)$ have the corresponding mean ± std values, 3.0233 ± 1.5694, 1.5999 ± 0.5582, and 2.0446 ± 0.6714.

### 8.4. Transformation from P-Adic Coordinates to Real Space-Time Coordinates

In order to understand the relation between the p-adic coordinates ${\tilde{z}}_{i}$ and the real space-time *t*, *x*, and *y* coordinates split the p-adic coordinates ${\tilde{z}}_{i}$ arbitrarily into three p-adic ${\tilde{t}}_{p}$, ${\tilde{x}}_{p}$, and $\tilde{y}$_p_ coordinates.

Each ${\tilde{z}}_{i}$ p-adic expansion was split in this manner:
$\tilde{x}$_p_ = 2 in the power of the last place of one digit of the p-adic expansion$\tilde{y}$_p_ = 2 in the power of the second last place of one digit of the p-adic expansion$\tilde{t}$_p_ = sum of all two places of the first digit until the second last.

For example:
$${\tilde{z}}_{i}=0\;0\;0\;1\;0\;1\;1\;1\;{\tilde{x}}_{i}=0\;0\;0\;00\;0\;0\;1\;{\tilde{y}}_{i}=0\;0\;0\;00\;0\;10\;{\tilde{t}}_{i}=0\;0\;0\;10\;1\;0\;0$$
we then could calculate a three-dimensional p-adic path interval as
$$\Delta path_{i\;padic}=ds_{{\tilde{t}}_{i,i+1}}+ds_{{\tilde{x}}_{i,i+1}}+ds_{{\tilde{y}}_{i,i+1}}$$

The real *t x y* path interval as
$$\Delta path_{i}={\left(x_{i}-x_{i+1}\right)}^{2}+{\left(y_{i}-y_{i+1}\right)}^{2}+{\left(t_{i}-t_{i+1}\right)}^{2}$$

The accumulated summation of $\Delta path_{i\;padic}\;and\;\Delta path_{i}$ was then fitted with either a linear function *y* = *bx* or power law *y* = *dx^m^* and the best fitted model was taken as a function between p-adic coordinates to real space-time coordinates. Figure 7A shows the values of m and b parameters that resulted from the fitting to power law or linear law. Figure 7B shows the CDFs of R^2^ values of all geodesics in each universe. Above 91.97 and 94% (in the corresponding 1–3 universes’ geodesics) have the best fit to the linear law with very high R^2^ values.

## 9. Concluding Remarks

We hope that this paper is a step towards the realization of Wheeler’s program of the information reconstruction of physics. As he wrote [4]: “It from bit symbolizes the idea that every item of the physical world has at bottom—at a very deep bottom, in most instances—an immaterial source and explanation; that what we call reality arises in the last analysis from the posing of yes-no questions and the registering of equipment-evoked responses; in short, that all things physical are information-theoretic in origin and this is a participatory universe”. We essentially modified Wheeler’s program: the bit-structure expresses not digitalization of data, but rather hierarchic relations hidden in physical events occurring in the universe.

DH reconstruction is based on the exclusion of objects located in space-time modelled with real continuum form physics. Following Wheeler [4], Rovelly [8], and Smolin [6], we consider physics as a theory of events not physical systems. Events are represented by information strings of zeros and ones, i.e., branches of a dendrogram. This is the epistemic model; transition to the ontic model is done straightforwardly via consideration of infinite trees. Here p-adic numbers arise. A p-adic universe preserves the information structure: the ontic events are encoded by infinite p-adic strings of zeros and ones in line with Wheeler’s project. Topologically, this information universe is very exotic and matches Bohm’s image of implicate order [32].

The DH theory also matches Mach’s principle. The universal potential *V* is nonlocal and determined by the topology of the dendrogram as a whole. In DH theory, such a *V* plays the role of the universal potential determining all elements of theory including analogues of the fundamental constants of nature. These analogues are dendrogram-dependent, *h* = *h*(*D*), *c* = *c*(*D*), and *G* = *G*(*D*). We found that (amazingly) the log-fractions of these *D* quantities (expressing information about them) are consistent with the log-fractions for the corresponding physical constants. This result was obtained for the special GR model: the Schwarzschild metric in a neighborhood of a black hole. In DH theory, a collection of geodesics contains (indirectly) information about the basic physical constants.

We did not try to quantize GR. We unified QM and GR through a new mathematical representation based on dendrograms (at the epistemic level) and p-adic numbers (ontic level). Quantum systems are represented by simple dendrograms and classical by complex ones. In this framework, the quantum–classical boundary is not sharp. The main characteristic of the quantum-like ensembles of dendrograms (We follow the statistical interpretation of quantum mechanics [33]. By this interpretation, quantum states are mathematical symbols encoding the ensembles of identically (and in real experiments, similarly) prepared quantum systems. However, DH theory excludes systems and it operates solely with events. Thus, the statistical interpretation is applied to ensembles of events. The ensembles of dendrograms whose members have a high degree of similarity are treated as quantum-like ensembles. Typically, such dendrograms should be simple: it is difficult to realize an ensemble of complex dendrograms (with long branches) whose individuals are very similar. “Quantumness” is checked with various statistical tests, e.g., the CHSH test (see [2])) is their simplicity; geometrically, these are trees with short branches. The ontic p-adic universe that is geometrically described by the infinite tree is classical. Thus, “quantumness” appears only at the level of observation.

Our main mathematical tool was numerical simulation based on the application of hierarchic clustering algorithms and construction of dendrograms (finite trees) from geodesics corresponding to metric tensors of GR (in this paper the Schwarzschild metric) (As was mentioned in the introduction, the generation of tree-like representation with hierarchic clustering algorithms is like the generation of the ultrametric structure in the theory of complex disordered systems (e.g., spin glasses). In the p-adic this was developed in works by Parisi and Sourlas [20] and Khrennikov and Kozyrev [34,35,36]). Like the works of Wheeler [4], Smolin [6], and Rovelli [8], space-time loses its fundamental role in DH theory. Smolin’s theory was based on the causal structure and not space-time. In DH theory, we consider the hierarchic structure instead of causal structure.

In contrast to Smolin [6], we emphasize that all events are always present in DH theory (this is more in accordance with Barbour’s [37] “always present events”) and does not need to appear by a dynamic process. We note that we do not need to postulate a casual structure, the fundamentality of time, momentum, and energy as in Smolin’s study. In contrast to Barbour, we do not require probabilities in the space phase to produce the apparent dynamics.

In future work, we will consider a variety of GR metrics and proceed to the basic GR phenomena.

## Figures and Tables

**Figure 1 entropy-24-00181-f001:**
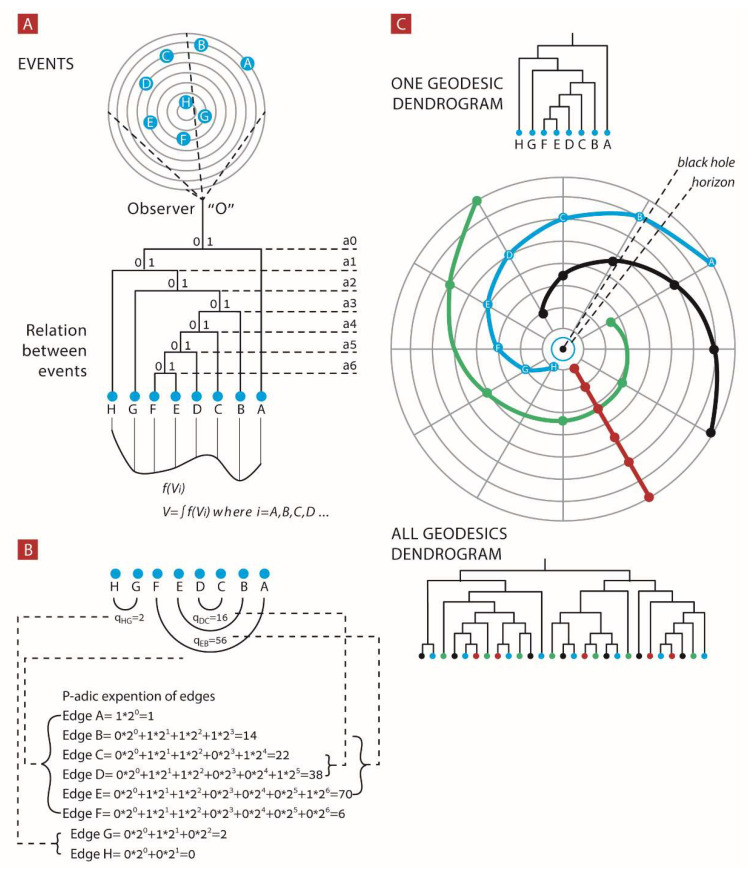
Relational observation of events. (**A**) Observer O discriminates events A–H and constructs an object, a dendrogram, which describes the relations between these events. (**B**) Each edge of the dendrogram is a binary string of 0s and 1s which can be represented as a finite p-adic expansion. Each edge summation of its finite p-adic expansion results in a natural number. Subtracting between two edges’ finite p-adic expansion results in “potential gap”—$q_{ij}$. (**C**) Dendrograms can be constructed by observing events from a single geodesic event or by observing events from several geodesics.

**Figure 2 entropy-24-00181-f002:**
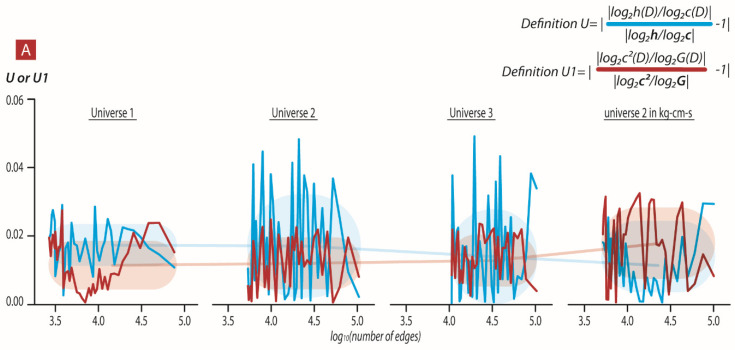
Comparison of the dendrogramic $\left|log_{2}h\left(D\right)\right|/\left|log_{2}c\left(D\right)\right|$ and $\left|log_{2}c^{2}\left(D\right)\right|/\left|log_{2}G\left(D\right)\right|$ ratios to the experimental $\left|log_{2}h\right|/\left|log_{2}c\right|$ and $\left|log_{2}c^{2}\right|/\left|log_{2}G\right|$ ratios. (**A**) Values of |($\left|log_{2}h\left(D\right)\right|/\left|log_{2}c\left(D\right)\right|)/\left(\left|log_{2}h\right|/\left|log_{2}c\right|\right)$ − 1| (blue line) upon increasing size of universal dendrogram of universes 1–3 where h and c are in kg-m-s units or kg-cm-s. Values of |($\left|log_{2}c^{2}\left(D\right)\right|/\left|log_{2}G\left(D\right)\right|)/\left(\left|log_{2}c^{2}\right|/\left|log_{2}G\right|\right)$-1| (red line) upon increasing size of universal dendrogram of universes 1–3 were G and c are in kg-m-s units (universes 1–3) or kg-cm-s units (universe 4). The mean ± std of |($\left|log_{2}h\left(D\right)\right|/\left|log_{2}c\left(D\right)\right|)/\left(\left|log_{2}h\right|/\left|log_{2}c\right|\right)$ − 1| for all coarse-grained universal dendrograms, in each universe, are represented as shaded blue with values 0.0115 ± 0.0071, 0.0121 ± 0.0075 and 0.0129 ± 0.0075 for universes 1–3 with h and c kg-m-s while universe 2 with h and c kg-cm-s results in 0.0178 ± 0.0104. The mean ± std of |($\left|log_{2}c^{2}\left(D\right)\right|/\left|log_{2}G\left(D\right)\right|)/\left(\left|log_{2}c^{2}\right|/\left|log_{2}G\right|\right)$ − 1| for all coarse-grained universal dendrograms, in each universe, are represented as shaded blue with values 0.0174 ± 0.0061, 0.0171 ± 0.0151, and 0.014 ± 0.014 for universes 1–3 with G and c kg-m-s while universe 2 with G and c kg-cm-s results in 0.0115 ± 0.0078. (**B**) Values of the free parameter $2^{\alpha }=1/\tau^{2}$ that results in better correpondance between of $\left|log_{2}c^{2}\left(D\right)\right|/\left|log_{2}G\left(D\right)\right|$ and $\left|log_{2}c^{2}\right|/\left|log_{2}G\right|$ for each universe,1–3, upon universal dendrogram size and topology. (**C**) The mean ± std of all |($\left|log_{2}h\left(D\right)\right|/\left|log_{2}c\left(D\right)\right|)/\left(\left|log_{2}h\right|/\left|log_{2}c\right|\right)$ − 1| where D is constructed for each geodesic with coarse-graining factor 20-1 for universes 1–3. (**D**) The mean ± std of all $|\left(\left|log_{2}c^{2}\left(D\right)\right|/\left|log_{2}G\left(D\right)\right|\right)/\left(\left|log_{2}c^{2}\right|/\left|log_{2}G\right|\right)$ − 1| where D is constructed for each geodesic with coarse-graining factor 20-1 for universes 1–3.

**Figure 3 entropy-24-00181-f003:**
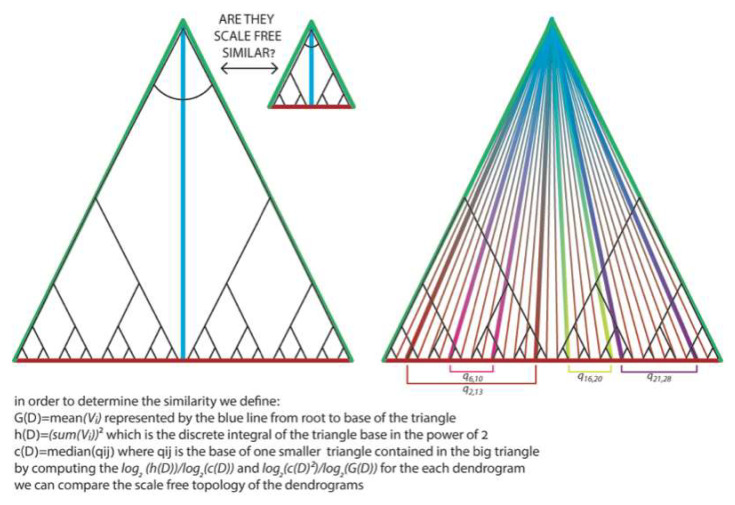
Heuristical geometrical description of the dendrogramic properties $h\left(D\right),\;c\left(D\right),$ and $G\left(D\right)$.

**Figure 4 entropy-24-00181-f004:**
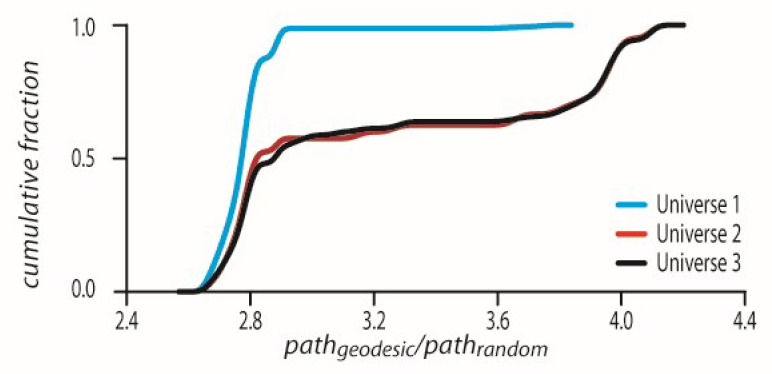
The p-adic path on ${\tilde{z}}_{i}$ coordinates is p-adically maximal.

**Figure 5 entropy-24-00181-f005:**
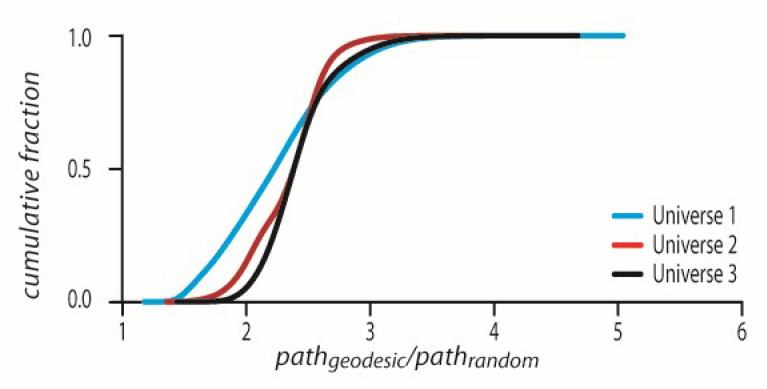
The p-adic path on ${\tilde{z}}_{i}$ coordinates where ${\tilde{z}}_{i}$ is constracted from each geodesic’s $V_{i}$ embaded in a dendrogram constructed from all geodesics and is p-adically maximal. The CDFs showing all ratio $path_{geodesic}/path_{random}$ values are above 1 for every geodesic in all universes 1–3.

**Figure 6 entropy-24-00181-f006:**
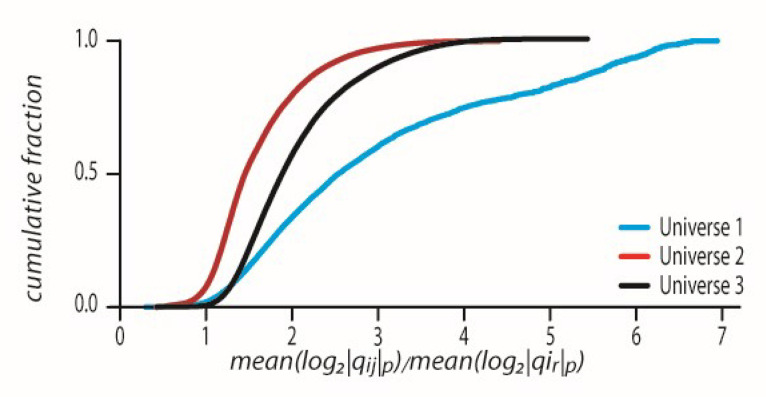
Events from one geodesic cannot move to another geodesic as the $q_{ir}$ values needed are bigger than the average potential gap inside a geodesic. The CDFs of all $mean\left(log_{2}|q_{ij}{|}_{p}\right)/mean\left(log_{2}|q_{ir}{|}_{p}\right)$ values where $q_{ij}$ and $q_{ir}$ are defined in Section 3 in each universe 1–3.

**Figure 7 entropy-24-00181-f007:**
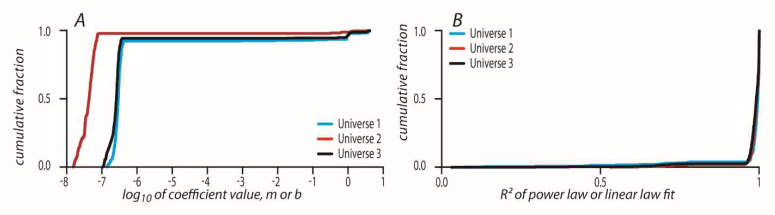
(**A**) Fitting parameters for the correspondence between the 3D p-adic path of geodesic and the real 2 + 1 geodesic path. Fitting with linear or power laws in each universe 1–3 (blue, orange, and yellow CDFs). (**B**) CDFs of R-squared values of the fittings with linear or power laws of the correspondence between the 3D p-adic path of geodesic and the real 2 + 1 geodesic path in each universe 1–3.

## Data Availability

Not applicable.

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
