# Peer review of "Towards Unification of General Relativity and Quantum Theory: Dendrogram Representation of the Event-Universe"

_entropy, 2022, doi:10.3390/e24020181_

Round 1

Reviewer 1 Report

In the present paper, the authors propose a model of fundamental physical objects based on ultrametric spaces, namely p-adic numbers, with the aim to conceptually 
start unifying the General Realtivity and Quantum Mechanics. This model is based on previous works in which the Dendrogramic Holography, the basic unifying theoretical structure, was proposed by the authors.

These are my comments on the paper:

1. The conceptual construction of dendrogram universe is interesting. However, the presentation of Dendrogramic Holography is very sketchy, at most, and it is 
relegated to previous papers, ref. 1 and 2. Since this is a very recent and unfamiliar construct, one concise presentation of it should be added either in the introduction or as an appendix.

2. Some inexact statements should be revised, e. g. line 85 on page 2: "This geometry can be represented by its geodesics." What exactly do the authors mean by that? The geodesics are solutions of the geodesic equation in a given geometry. 
In which way these solutions can be used to distinguish geometries? Also, on line 89 "Our basic tool is numerical simulation." What is the numerical simulation used for? 
Since the authors aim at representing the basic structure of the quantum and gravitational phenomena, they should specify what exactly is numerically simulated from the quantum and gravitational theories. Another example is on page 3, line 124 
where the authors state "In DH-gravity,..." without explaining what this theory of gravity is, its basic laws, etc. I suggest whenever such ambiguous formulations appear in the text, either be removed or be given a more exact form. 

3. On page 6, the authors present the construction of dendrograms by an observer. After consulting ref. 1, I was able to have a better grasp on the process. However, there are still some questions that the authors should address. Firstly, it is not
clear how the dendrograms are related to the local coordinates systems. Section 6 "Dendrogram-representation of geometry around Schwarzschild black hole" gives an idea of
the process in two dimensions but it is still very schematic. Secondly, it is not clear how the dendrograms are to be related to different local coordinates, e. g. overlapping charts of space-time manifold, covariant structure even in flat space-time, etc. Thirdly, it is not clear how the clustering batches of 
the geodesics for black holes are to be constructed in four dimensions. Fourthly, it is not clear what is the relation between the dendrograms and the causal structure of space-time, which is a quite important issue in order to guarantee the construction of 
a dendrogram based on physical events.

4. In general, from the information given in the paper and in references [1] and [2] it is not possible to either validate or invalidate the results presented by authors. I suggest them giving more details on their theory, if not in the analytical form which seems to be unavailable yet, at least of its algorithm and its numerical implementation for geodesics, dimensions, etc. For example, how the clustering batch depends on the black hole mass. That could be added as an appendix.

5. In section 5, it is claimed that the dendrograms constructed from geodesics for Schwarzschild metric in the neighborhood of a black hole have coincidental numerical objects (the $\log_2$ fractions of the universal constants) with the same objects 
constructed from measured data. However, without more details on the algorithm used in the paper, the value of constants considered in it (e. g. black hole mass dependence) it is impossible to verify this claim or to understand it. 
The authors give the general explanation on the page 8 line 322 "dendrograms are generated by the clustering algorithm from batches of geodesics for Schwarzschild metric in the 
neighborhood of a black hole". This sentence is very general , see the point 5 above.

6. Nowhere in the paper, with the exception of the very general motivation that the p-adic numbers could be used to describe the quantum properties of matter given in the introduction, are the results related to the quantum field theory or quantum mechanics. Since the claim of unification is strongly stated at the beginning, either that should be removed or the appearance of the quantum properties, even in a rudimentary form, should
be shown. 

7. There are some important references missing on the subject of the p-adic numbers and gravity emergence, quantum field theory and holography. I suggest consulting the following references and comparing the approach from this paper to the approaches given there:

Emergent Einstein Equation in p-adic Conformal Field Theory Tensor Networks, Lin Chen, Xirong Liu, and Ling-Yan Hung, 
Phys. Rev. Lett. 127, 221602 (2021).

p-adic CFT is a holographic tensor network, Ling-Yan Hung, Wei Li and Charles M. Melby-Thompson , JHEP (2019) 170.

Edge length dynamics on graphs with applications to p-adic AdS/CFT, Steven S. Gubser, Matthew Heydeman, Christian Jepsen, Matilde Marcolli, Sarthak Parikh, Ingmar Saberi, Bogdan Stoica, Brian Trundy, JHEP 06 (2017) 157. 

Tensor networks, p-adic fields, and algebraic curves: arithmetic and the AdS_3/CFT_2 correspondence, Matthew Heydeman, Matilde Marcolli, Ingmar Saberi, Bogdan Stoica,
Adv.Theor.Math.Phys. 22 (2018).

p-adic AdS/CFT, Steven S. Gubser, Johannes Knaute, Sarthak Parikh, Andreas Samberg, Przemek Witaszczyk,
Commun.Math.Phys. 352 (2017) 3, 1019-1059.

In conclusion, the paper presented here has interesting claims. An improvement in the presentation, namely clarifying ambiguous or too general statements and typesetting text
and formulas as well as giving details on how the proposed results are obtained, are in order. Once these issues are addressed, I recommend the paper for publication.

Author Response

Dear Reviewer,

We would like to thank you for the constructive critical comments and suggestions for improving the paper. We worked seriously on them, for new material see Figure 1, sections 1.5, 6, and Appendix 2A. We also thank you for informing us about new interesting research combining p-adics, holography, and quantum gravity. We were unaware about them; we shall study them in future in more detail.

1.the conceptual construction of dendrogram universe is interesting. However, the presentation of Dendrogramic Holography is very sketchy, at most, and it is  relegated to previous papers, ref. 1 and  Since this is a very recent and unfamiliar construct, one concise presentation of it should be added either in the introduction or as an appendix.

Authors: We created Figure 1 summarising in the compact form our approach. We hope that it will be helpful for the readers.

  1. Some inexact statements should be revised, e. g. line 85 on page 2: "This geometry can be represented by its geodesics." What exactly do the authors mean by that? The geodesics are solutions of the geodesic equation in a given geometry.  In which way these solutions can be used to distinguish geometries?

Authors: Now we wrote: “In GR, space-time geometry is determined by a metric tensor g=(g_{ij}). This geometry can be extracted from its corresponding geodesic equation and its solutions the geodesics.”

Also, on line 89 "Our basic tool is numerical simulation." What is the numerical simulation used for? 
Since the authors aim at representing the basic structure of the quantum and gravitational phenomena, they should specify what exactly is numerically simulated from the quantum and gravitational theories.

Authors: the above statement was replaced by the extraction of data from numerical simulation of photon trajectories on a space-time manifold

 Another example is on page 3, line 124  where the authors state "In DH-gravity,..." without explaining what this theory of gravity is, its basic laws, etc. I suggest whenever such ambiguous formulations appear in the text, either be removed or be given a more exact form. 

Authors: Yes, we agree, we deleted this expression and throughout the paper we use the terminology “DH-theory”.

  1. On page 6, the authors present the construction of dendrograms by an observer. After consulting ref. 1, I was able to have a better grasp on the process. However, there are still some questions that the authors should address. Firstly, it is not clear how the dendrograms are related to the local coordinates systems. Section 6 "Dendrogram-representation of geometry around Schwarzschild black hole" gives an idea of the process in two dimensions but it is still very schematic.

Authors: We worked on clarification, see Figure 1, section 6, and appendix 2A.

Secondly, it is not clear how the dendrograms are to be related to different local coordinates, e. g. overlapping charts of space-time manifold, covariant structure even in flat space-time, etc.

Authors: see section 6. The geodesics which where simulated by the Schwarzschild metric already include information on connection of different local coordinates through tensor networks as well as information on the Schwarzschild black hole mass. Thus dendrograms constracted from these geodesics events have encoded in them already the different local coordinates.

Thirdly, it is not clear how the clustering batches of the geodesics for black holes are to be constructed in four dimensions.

Authors: the same technique works in any dimension. We added in appendix 2A the following : In principle  can have any number of coordinate or data features vector. Thus any number of dimensions of any coordinate system  can be treated with exactly the proceadures outlined here.

Fourthly, it is not clear what is the relation between the dendrograms and the causal structure of space-time, which is a quite important issue in order to guarantee the construction of 
a dendrogram based on physical events.

Authors: We emphasized that the casual structure is an apparent casual structure which comes about by maximizing the action principle. We do not have a casual structure in DH-theory -- it is an emergent property. See section 6. We also added in section 8. We emphasize that the casual structure on the static dendrogram is an emergent property. This casual structure emerges upon maximizing the above action principle   according to the dendrogramic structure (as will be demonstrated below in sections 8.1-8.4). this dendrogramic casual structure is in agreement with  the “real world” casual structure. More over, although different coordinates systems, metrics and linkage algorithms will produce different dendrogramic casual structure the agreement with the “real world” casual structure will be preserved.

More over all of section  8 (sections 8.1-8.4) is devoted to describe   numerically how the casual structure emerges in our simulated manifold which is described by null-geodesics created by a black hole.

  1. In general, from the information given in the paper and in references [1] and [2] it is not possible to either validate or invalidate the results presented by authors. I suggest them giving more details on their theory, if not in the analytical form which seems to be unavailable yet, at least of its algorithm and its numerical implementation for geodesics, dimensions, etc. For example, how the clustering batch depends on the black hole mass. That could be added as an appendix.

Authors: The black hole mass is already manifested in the space-time structure which results in null-geodesics. So, yes its mass is encoded in the real space-time structure and then in the p-adic dendrogram. See section 6. We also added this comment: we have to point that the space time geometry and the events it encompasses defines the black hole mass. 

  1. In section 5, it is claimed that the dendrograms constructed from geodesics for Schwarzschild metric in the neighborhood of a black hole have coincidental numerical objects (the $\log_2$ fractions of the universal constants) with the same objects 
    constructed from measured data. However, without more details on the algorithm used in the paper, the value of constants considered in it (e. g. black hole mass dependence) it is impossible to verify this claim or to understand it. The authors give the general explanation on the page 8 line 322 "dendrograms are generated by the clustering algorithm from batches of geodesics for Schwarzschild metric in the  neighborhood of a black hole". This sentence is very general , see the point 5 above.

Authors:see again section 6 and appendix 2A.

  1. Nowhere in the paper, with the exception of the very general motivation that the p-adic numbers could be used to describe the quantum properties of matter given in the introduction, are the results related to the quantum field theory or quantum mechanics. Since the claim of unification is strongly stated at the beginning, either that should be removed or the appearance of the quantum properties, even in a rudimentary form, should be shown. 

Authors: This is the complex issue, and we are at the very beginning of the new approach to unification of quantum and classical physics. We essentially extended section 1.5, including possible experimental tests.

  1. There are some important references missing on the subject of the p-adic numbers and gravity emergence, quantum field theory and holography. I suggest consulting the following references and comparing the approach from this paper to the approaches given there:

Emergent Einstein Equation in p-adic Conformal Field Theory Tensor Networks, Lin Chen, Xirong Liu, and Ling-Yan Hung, 
Phys. Rev. Lett. 127, 221602 (2021).

p-adic CFT is a holographic tensor network, Ling-Yan Hung, Wei Li and Charles M. Melby-Thompson , JHEP (2019) 170.

Edge length dynamics on graphs with applications to p-adic AdS/CFT, Steven S. Gubser, Matthew Heydeman, Christian Jepsen, Matilde Marcolli, Sarthak Parikh, Ingmar Saberi, Bogdan Stoica, Brian Trundy, JHEP 06 (2017) 157. 

Tensor networks, p-adic fields, and algebraic curves: arithmetic and the AdS_3/CFT_2 correspondence, Matthew Heydeman, Matilde Marcolli, Ingmar Saberi, Bogdan Stoica,
Adv.Theor.Math.Phys. 22 (2018).

p-adic AdS/CFT, Steven S. Gubser, Johannes Knaute, Sarthak Parikh, Andreas Samberg, Przemek Witaszczyk,
Commun.Math.Phys. 352 (2017) 3, 1019-1059.

Authors: we thank the reviewer for pointing us to these articles. we added these articles and  suggested possible link beyween them and   our study .

In order to simulate these trajectories different local coordinates need to be connected by tensor networks.

We note that tensor networks in the proximity of a black hole have already been linked to p-adic numbers and p-adic trees with some analytical solutions that indicate the   emergence of gravity, quantum field theory and holography[27-31 ].

And also

The geodesics which where simulated by the Schwarzschild metric already include information on connection of different local coordinates through tensor networks as well as information on the Schwarzschild black hole mass.

In conclusion, the paper presented here has interesting claims. An improvement in the presentation, namely clarifying ambiguous or too general statements and typesetting text
and formulas as well as giving details on how the proposed results are obtained, are in order. Once these issues are addressed, I recommend the paper for publication.

Authors: Thank you for your support of our novel approach.

Reviewer 2 Report

Please see the attached report.

Author Response

Thank you for your support of our novel approach.

Round 2

Reviewer 1 Report

The manuscript has been improved considerably in the present version, therefore I recommend it for publication as it is.